# Checkpoint Inhibition in Bladder Cancer: Clinical Expectations, Current Evidence, and Proposal of Future Strategies Based on a Tumor-Specific Immunobiological Approach

**DOI:** 10.3390/cancers13236016

**Published:** 2021-11-29

**Authors:** Mariangela Mancini, Marialaura Righetto, Elfriede Noessner

**Affiliations:** 1Urological Clinic, University Hospital of Padova, 35121 Padova, Italy; marialaurarighetto@gmail.com; 2Department of Surgical, Oncological and Gastroenterological Sciences, University of Padova, 35121 Padova, Italy; 3Helmholtz Center Munich, Immunoanalytics Research Group-Tissue Control of Immunocytes, 81377 Munich, Germany; noessner@helmholtz-muenchen.de

**Keywords:** checkpoint inhibition, bladder cancer, immunotherapy biomarkers, cancer immunoprofiling, BCG failure, randomized clinical trials, urological research

## Abstract

**Simple Summary:**

In contrast with other strategies, immunotherapy is a treatment aimed at empowering the patient’s immune system in order to increase immunity and the response against cancer. Recently, a new class of drugs, immune checkpoint inhibitors, has shown potential in increasing treatment chances for patients with bladder cancers, improving their survival. However, predicting the response to immune checkpoint inhibition is important, since only a group of patients develop a good response. Biomarkers to predict the response to checkpoint inhibition must identify tumors’ and patients’ specific profiles. This study reviews the current knowledge on this most relevant clinical topic, focusing on bladder cancer, going from basic science to ongoing clinical trials and available clinical evidence. Finally, a critical analysis of published data is provided, and an original panel of biomarkers, able to select the right patients for treatments, based on patient-specific immune profiling, is proposed.

**Abstract:**

In contrast with other strategies, immunotherapy is the only treatment aimed at empowering the immune system to increase the response against tumor growth. Immunotherapy has a role in the treatment of bladder cancer (BC) due to these tumors’ high tumor mutational burden (TMB) and mostly prominent immune infiltrate. The therapy or combination has to be adjusted to the tumor’s immunobiology. Recently, a new class of immunotherapeutic agents, immune checkpoint inhibitors (ICI), has shown potential in increasing treatment chances for patients with genitourinary cancers, improving their oncological outcomes. The clinical efficacy of ICI has been shown in both the first-line treatment of cisplatin-ineligible patients, with programmed death ligand 1 (PD-L1)-positive tumors (atezolizumab, pembrolizumab), and in second-line settings, for progression after platinum-based chemotherapy (atezolizumab, pembrolizumab, and nivolumab for FDA and EMA; durvalumab and avelumab for FDA alone). Predicting the response to ICI is important since only a subset of patients undergoing ICI therapy develop a concrete and lasting response. Most of the patients require a different therapy or therapy combination to achieve tumor control. The cancer immunity cycle provides a conceptual framework to assist therapy selection. Biomarkers to predict response to ICI must identify where the cancer immunity cycle is disrupted. We reviewed the current knowledge on ICI treatment in BC, going from basic science to current data and available clinical evidence. Secondly, a critical analysis of published data is provided, and an original panel of biomarkers able to predict response to ICI treatment, based on tumor-specific immune profiling, is proposed.

## 1. Background: Bladder Cancer and the Promises of Immunotherapy

Bladder cancer (BC) is the seventh most commonly diagnosed cancer in males worldwide, and the eleventh when considering both genders [1]. The worldwide age-standardized incidence rate (per 100,000 person/years) is 9.0 for men and 2.2 for women. In 2018, nearly 550,000 new cases were diagnosed worldwide, with 200,000 deaths [1].

The most common histological type of BC is urothelial carcinoma. Moreover, about 75% of patients with BC present with a non-muscle-invasive disease (non-muscle-invasive bladder cancer—NMIBC) confined to the mucosa (stage Ta; carcinoma in situ—CIS) or submucosa (stage T1). NMIBC is classified in different risk groups according to different prognostic factors [2], and it has different recurrence rates that require several endoscopic transurethral treatments. On the other hand, muscle-invasive and metastatic BC need multimodal strategies, including surgery and chemotherapy, in neoadjuvant, adjuvant, or palliative settings.

Ten to fifteen percent of patients with muscle-invasive BC are already metastatic at diagnosis [3]. Moreover, approximately 50% of patients with muscle-invasive non-metastatic BC will relapse after radical cystectomy, mostly with distant metastases (30% local recurrence, 70% distant metastases).

Several strategies have been tested to improve the disease-free survival (DFS) and overall survival (OS) of these patients, testing the role of chemotherapy and radiotherapy in neoadjuvant and adjuvant settings. At present, locally advanced disease is preferentially treated with neoadjuvant cisplatin-based chemotherapy, which is able to achieve an 8% 5-year absolute improvement in OS [4]. Adjuvant treatment, on the other hand, remains a valid option for high-risk diseases [5]. Despite several efforts to develop a more effective pre- and/or postoperative treatment, the OS for metastatic BC patients who received platinum-based chemotherapy is estimated to be 12–14 months, and it is reduced to <7 months in patients with a relapsing disease [6]. Moreover, up to 50% of patients with metastatic BC are ineligible for cisplatin-based chemotherapy [7] (patients with at least one of the following criteria: performance status >1; glomerular filtration rate ≤ 60 mL/min; grade ≥2 audiometric loss; peripheral neuropathy; and New York Heart Association class III heart failure).

In contrast with other strategies, immunotherapy is the only treatment aimed at empowering the immune system to increase the response against tumor growth. Recently, a new class of immunotherapeutic agents, immune checkpoint inhibitors (ICI), has shown potential in increasing treatment chances for patients with genitourinary cancers, improving their oncological outcomes [8].

ICI are approved for use in metastatic BC by both the US Food and Drug Administration (FDA) and the European Medicines Agency (EMA).

Clinical efficacy of ICI has been shown in both the first-line treatment of cisplatin-ineligible patients, with programmed death ligand 1 (PD-L1)-positive tumors (atezolizumab, pembrolizumab), and in second-line settings, for progression after platinum-based chemotherapy (atezolizumab, pembrolizumab, and nivolumab for FDA and EMA; durvalumab and avelumab FDA alone) [9,10,11,12,13,14,15,16].

The aim of this paper is to review the current knowledge on ICI treatment in BC, going from basic science to current data and available clinical evidence. Secondly, a critical analysis of published data is provided, and a panel of biomarkers able to predict response to ICI treatment, based on tumor-specific immune profiling, is proposed.

## 2. Immune Checkpoint Inhibitors for Cancer Treatment

### 2.1. Tumor Immune Escape beyond the “Checkpoint Brakes”

Checkpoint inhibition has transformed cancer therapy, achieving remarkable responses across many cancer types. Yet, only fractions of patients are responding or responding long term [17,18]. For urothelial cancer, the initial excitement that checkpoint inhibition may change the management of the disease gave way to disappointment as several phase III trials failed to show improvement over chemotherapy, and clinical trials using seemingly similar immunotherapeutic agents showed inconsistent results [19,20,21,22]. However, there is still reason for excitement about immunotherapy’s potential, and maximizing patient benefit seems possible. For progress to occur, attention has to be directed to the immunobiology of urothelial cancer, and therapies need to be adjusted towards the tumor’s individual display of immune responsiveness and resistance features. As outlined in the following paragraph describing the cancer immunity cycle ([23], Figure 1), ICI, also defined as “releasing the T cell’s brakes”, is only one aspect within the process of inducing and maintaining an antitumor response. Therefore, ICI cannot achieve a response in all patients.

For an immune response to be successful, a series of events have to elapse in a stepwise manner, schematically illustrated by the cancer immunity cycle ([23], Figure 1). In the first step, tumor cells have to express and release antigens ①, which are taken up and processed by antigen-presenting cells, i.e., dendritic cells (DCs) ②. DCs need to mature and migrate to lymph nodes, where they present the antigens (as peptide/MHC complexes) to naïve T cells. T cells start to proliferate and differentiate into effector cells, i.e., cytotoxic T cells ③, which leave the lymph node and migrate to peripheral tissues ④ in search for the antigen that has caused their activation. T cells need to find the tumor site, extravasate, and invade the tissue ⑤, where they can recognize the tumor cells, if the antigen is presented on the tumor cell surface through the MHC molecules ⑥. Elimination of the tumor cells then occurs ⑦. With tumor cell killing, new antigens are released that can be captured by DCs, starting the cycle anew with new effector T cells being generated and more tumor cells being killed. “Pushing and pulling” is required to keep the cycle turning [24]. If the process is effective, all antigens are cleared and the tissue returns to homeostasis.

However, the T cell response and tumor cell elimination will be ineffective, allowing tumors to develop and progress, if the cycle is interrupted at any of these steps. The checkpoint molecules cytotoxic T lymphocyte-associated protein 4 (CTLA-4) and programmed cell death protein 1 (PD-1)/programmed death ligand 1 (PD-L1) are active at the steps of T cell priming (step ③) and elimination (step ⑦), respectively. These steps are physiologically crucial to prevent detrimental over-reactivity of the immune response, preventing the development of too many T cells and autoreactive T cells (step ③), and limiting the killing activity to avoid collateral tissue damage (step ⑦).

Tumors take control of these physiological “brakes”, diminishing the immune response and allowing tumor cell escape from immune destruction. However, beyond the CTLA-4 and PD-1/L1 brakes, tumors subvert the immune system by many other ways [25]. Among them, the major hindrance of immune-mediated tumor killing is the hostile tumor microenvironment (TME). This is generated by the tumor cells through their extensive metabolism, depleting nutrients such as glucose or glutamine, which the T cells require for activity, and generating T cell inhibitory metabolites, such as lactate and acidosis. Through secretion of chemokines, tumor cells attract regulatory T cells (Tregs) and macrophages, which may be polarized in the TME towards myeloid suppressor cells, MDSCs, and tumor-associated macrophages (TAM). VEGF, IL-10, or prostaglandins (PGE_2_) disturb DC biology by inhibiting their capacity to migrate and present antigens to the T cells for priming in the lymph nodes. TGF-ß contributes to T cell exclusion [26] and loss of cytotoxic proteins in CD8 T cells [27].

Current checkpoint inhibitors correct deficits at two steps: anti-CTLA-4 reagents (i.e., ipilimumab, tremelimumab, which are both used in the treatment of BC) act at the priming phase ③ by inhibiting the negative signal of CTLA-4 resulting from the binding of CTLA-4 to B7.1 and B7.2 expressed on B lymphocytes, DCs, and macrophages; anti-PD-1 therapeutics (i.e., nivolumab, pembrolizumab) or anti-PD-L1 (i.e., atezolizumab, avelumab, durvalumab) act at the elimination phase ⑦. Anti-CTLA-4 and anti-PD-1/L1 operate at different points of the cycle through different mechanisms. Thus, combining both can improve response rates, which is seen in clinical trial results [28].

However, the cycle can be interrupted at many other steps. In these situations, checkpoint inhibitors, even if used in combination, do not repair the cycle, and the antitumor response will be limited or not occur at all. One example of many is TGF-ß, which is highly expressed by mUC, attenuating the tumor response to PD-L1 blockade [26].

### 2.2. Biomarkers for Therapy Selection: Considerations from Immunobiology

Predicting the response to immune checkpoint blockade is important since only a subset of patients undergoing ICI develop a concrete and lasting response [18]. Most of the patients require a different therapy or therapy combination to achieve tumor control.

Biomarkers to predict response to ICI must identify where the cancer immunity cycle is disrupted. Currently, FDA-approved biomarkers are PD-L1 expression and microsatellite instability-high (MSI-H)/mismatch repair deficiency (dMMR), tumor mutational load (TMB), gene expression profiles, and tumor-infiltrating lymphocytes (TIL) [29].

From an immunology standpoint, all of them have value, but none of them alone have, and cannot have, sufficient predictive power since each one represents only part of the complexity and individuality of the tumor biology and immune response.

#### 2.2.1. PD-1/PDL-1

PD-L1 detection by IHC is FDA approved and has demonstrated prognostic value in some situations. However, it is mostly unsatisfactory to predict treatment efficacy with patients failing to respond despite PD-L1-positive tumors, and response is observed with patients despite the absence of PD-L1 in the TME [19,20,21,30].

Besides methodological limitations, including poor concordance among approved assays, differences in cut-offs, and multiple scoring systems defining PD-L1 protein expression on tumor and/or immune cells [19,20,29,31], the immunobiology of PD-L1 disqualifies this marker as a biomarker sufficient, by itself, for therapy response prediction and patient selection, outlined as follows:

(1) PD-L1 is just one inhibitory component in the TME among many others, including cytokines, suppressor cells, MHC loss, or the absence of T cells. Tumors that are devoid of T cells, do not express MHC or antigens, are infiltrated with inhibitory cells (Tregs, suppressive macrophages), or exhibit inhibitory mechanisms (lactic acidosis, glucose depletion, suppressive cytokines) in the TME cannot (fully) respond to ICI even if the PD-1/PD-L1 pathway inhibition is overcome (Figure 2).

(2) PD-L1 expression is dynamic and regulated by a variety of tumor cell-intrinsic pathways, i.e., oncogenic k-ras or myc, as well as extrinsic factors, such as cytokines (IFN-γ) and treatment regimens (radiation, BCG, chemotherapy) [32,33]. Thus, it is difficult to judge what the absence or presence of PD-L1 at a given time point really means [34]. An initially PD-L1-negative tumor that was unresponsive to anti-PD-1/L1 therapy may be PD-L1 positive after BCG treatment [35], and then may be responsive to checkpoint therapeutics. A single look at PD-L1 may mislead the treatment decision. However, an informed consideration of the PD-L1 expression pattern together with other markers (TMB, CD8, suppressor cells) and the spatial relationship of PD-L1 to CD8 expression could be of significant value. Moreover, distinguishing PD-L1 expression on tumor cells from expression on tumor-infiltrating mononuclear cells has shown predictive value of overall survival in urothelial cancer patients who developed metastasis [36].

#### 2.2.2. Tumor Mutational Burden (TMB)

Tumor neoantigens (i.e., mutated proteins) are optimal antigens for T cell recognition since they are foreign to the immune system [37,38]. Therefore, tumors with a high TMB are more likely to be targets for T cell attack. An association with immunotherapy response is conceivable, and some predictive capacity has been reported [39], variably across various tumor indications [19,20,21]. Yet, despite a high TMB, tumors can be resistant to the T cell response and ICI will be ineffective if the tumor is not infiltrated by T cells, if the tumor cells do not express MHC for antigen presentation [40], and/or if suppressor cells or metabolites are present in the TME that inhibit T cell activity. TMB is, thus, subordinate to the presence of T cells, the antigen presentation machinery, and the TME (Figure 3). Considering this, TMB cannot be a sufficient marker by itself for predicting response to checkpoint inhibition. The immunological shortcomings add to technical limitations, including a lack of harmonization of TMB tests across different platforms, the variability in cut-off definitions, and high technical requirements and costs, which make the application of TMB testing in the clinical routine difficult [41,42,43].

#### 2.2.3. Tumor-Infiltrating Immune Cells (TILs) and T Cells

TILs, as defined on H&E sections, have been added to the list of potential biomarkers. Considering TILs as biomarkers for response to checkpoint inhibition is plausibly based on the knowledge that the mechanism of checkpoint therapeutics is the restoration of T cell activity. The International Immuno-Oncology Biomarker Working Group has published guidelines for TIL counting in solid tumors, including UCB [44]. Yet, it is not established whether an H&E-based assessment can provide sufficient information for clinical decision making in the context of immunotherapy. One shortcoming of TIL counts on H&E is that it cannot differentiate lymphocyte subsets. Thus, TIL counts may include antitumor reactive effector lymphocytes (NK cells, CD8 or CD4 lymphocytes) but also immunosuppressive cells (Treg, TAM). High levels of TILs may therefore not correspond to high levels of active antitumor T cells, one possible explanation why some patients with high TILs do not show an improved prognosis. An improvement over TIL counts is immunohistology using CD3, CD4, and CD8 antibodies, which allow discriminating T cell subsets. Indeed, early trial results identified high intratumoral CD8 T cells, but not CD4 T cell numbers, as a correlate of response to anti-PD-1 therapy in various tumor types, including UC [45], and risk for recurrence [46,47]. NK cells are another type of cytotoxic lymphocyte with predictive value in some tumor types, including renal cell carcinoma [48]. In UC, the role of NK cells in prognosis and therapy response is unclear, with reports suggesting a positive contribution to the response to BCG therapy [49], but also an association with larger tumor size, i.e., poor prognosis, has been reported [50]. Better methods and markers for the detection and quantification of NK cell subtypes, which exhibit different functions, are warranted before this potentially important cell type can be integrated into clinical decision making.

Even though immune histology can discriminate immune cell subsets and identify the potentially valuable CD8 T cells, the knowledge on CD8 by itself is not sufficient to predict response to ICI since the “brakes”, released by checkpoint inhibitors, are only one part of immune escape that hinders the antitumor response. Additional mechanisms in the TME, including suppressor cells [51], inhibitory cytokines and metabolites, and lack of antigen presentation and MHC [40], may prevent T cell activity even though the brakes are released (Figure 3).

#### 2.2.4. Gene Expression Signatures and Genomic Mutations

Gene expression signatures are an assembly of gene transcripts that are associated with distinct states of the immune response and tumor biology. In the IMvigor 210/211 study of atezolizumab [15], a good response was seen for tumors that had a high T effector signature and no stromal signature. Low/no response rates were seen when the T effector signature was absent or a stromal signature was additionally present. In another study, a 25-gene IFN-γ signature was associated with response to nivolumab in mUC after platinum therapy [52]. Tang et al. [53] defined four immunotypes of BC, referred to as C1–C4. C2 had the highest signature for immune cell infiltration and an interferon response signature, while C4 exhibited a “desert”-like (non-inflamed) phenotype deprived of CD8+ cells. Patients with tumors showing signatures of high immune cell infiltration were more sensitive to anti-PD-1/L1 treatment than other subtypes [53].

Gene signatures provide deep insight into the tumor biology and show an association with treatment response. However, the technology is very complex and costly, making its wide application in clinical practice challenging.

Recent attempts utilized the knowledge of commonly observed genomic mutations to build risk scores to help predict therapy response. For BC, a mutation classifier including TP53 (tumor protein 53), PICK3CA (phosphatidylinositol-4,5-Bisphosphate 3-Kinase Catalytic Subunit Alpha), and ATM (ataxia-telangiectasia mutated) showed predictive value for response to ICI, but not to non-ICI therapies [54]. Patients in the low-risk group of the classifier (i.e., lower frequency of mutations) and with a good response to ICI had tumors with gene signatures indicative of a higher and more active immune infiltration. This finding is consistent with a previous observation that mutations in these proteins are associated with the infiltration of M2-polarized pro-tumorigenic macrophages [55].

## 3. Randomized Clinical Trials on Immunotherapy in Bladder Cancer

### 3.1. Endpoints in Clinical Trials

In the next paragraphs, we discuss clinical endpoints of RCTs on immunotherapy in BC. In addition to outcome measures of RCTs (OS, CSS, RFS), we selected two other outcome measures from the different RCTs: the pathologic complete response rate (pCRR) and the objective response rate (ORR). The pCRR is the pT0 finding at the time of radical cystectomy. The ORR, instead, is defined by the radiologic Response Evaluation Criteria in Solid Tumors version 1.1 (RECIST v1.1), which measures disease size at computed tomography scan or X-ray. The ORR can be differentiated into the percentage of patients whose disease decreased (partial response) and/or the percentage of patients whose disease disappears (complete response) after treatment.

### 3.2. Search Strategies, Study Selection, and Data Extraction

We performed a narrative review based on databases including PubMed, MEDLINE, Embase, and the Cochrane Library. We used for the search different combinations of the following keywords: “bladder cancer”, “urothelial carcinoma”, “non-muscle-invasive bladder cancer”, “muscle-invasive bladder cancer”, “metastatic bladder cancer”, “neoadjuvant immunotherapy”, “adjuvant immunotherapy”, “neoadjuvant checkpoint inhibitors”, “adjuvant checkpoint inhibitors”, “overall survival”, “cancer-specific survival”, and “recurrence-free survival”.

We included only the most recent results, published from January 2018 up to 31 July 2021. No language restriction was applied. Eligible studies were randomized controlled trials (RCT) only, reporting survival analysis (Kaplan–Meier plots) and Cox proportional multivariable models analyzing OS, recurrence-free survival (RFS), and cancer-specific survival (CSS). The references of previous meta-analyses were also screened for relevant results. Full-text articles and non-full-text articles only (i.e., conference abstracts) were included in this review.

We excluded RCTs on the ICI treatment for upper urinary tract urothelial carcinoma.

The authors screened articles independently and selected studies for inclusion. Discrepancies between the authors were resolved by discussion.

## 4. Non-Muscle-Invasive Bladder Cancer Unresponsive to BCG

NMIBC has high recurrence rates, up to 60%, within the first year of diagnosis [2]. Standard treatment for high-risk NMIBC (CIS, T1G1, high-grade Ta) involves transurethral resection of the tumor followed by intravesical Bacillus Calmette–Guérin (BCG) induction and maintenance therapy for up to 3 years [2]. Adjuvant therapy with BCG is currently used to prevent recurrence in these high-grade NMIBCs. Although 70% of patients with NMIBC achieve pCRR, recurrence and/or progression to muscle-invasive bladder cancer (MIBC) occurs in up to 80% and 50% of patients, respectively (BCG-unresponsive patients), the majority of them (80%) having recurrence within 1 year [56]. Previous studies reported that BCG stimulates TH1 response and the recruitment of CTL/NK cells.

The immune checkpoint receptor PD-1 is commonly expressed on activated T lymphocytes that modulate immune response. In particular, tumor cells may use the interaction between PD-1 and its ligands PD-L1 to escape immune-mediated cytotoxicity [57]. An altered PD-1 pathway has been observed in NMIBC recurrence and progression and in BCG resistance.

Moreover, BCG raises PD-L1 expression, with 20-fold higher PD-L1 expression in BCG-unresponsive patients. Therefore, targeting the PD-1/PD-L1 pathway has emerged as a potential therapeutic option for NMIBCs. Recently, pembrolizumab obtained FDA approval for the treatment of BCG-unresponsive patients who refused or were ineligible for radical cystectomy, based on the preliminary results of the Keynote-057 phase II trial [58,59]. The trial enrolled BCG-unresponsive patients to receive pembrolizumab (200 mg) every three weeks, until life-threatening toxicity, persistent/recurrent high-risk NMIBC, progression to MIBC, or up to 24 months of treatment. Preliminary data showed a complete response after 3 months of treatment in 40.2% of patients; moreover, 72.5% of these patients maintained it at a median follow-up of 14 months (80.2% of them had a complete response that exceeded 6 months). However, it must be emphasized that the Keynote-057 trial is still enrolling patients without carcinoma in situ (CIS), while data on the efficacy of pembrolizumab are available only for a cohort with carcinoma in situ (CIS) with/without papillary disease.

Promising results of a phase I trial of intravesical BCG combined with pembrolizumab in high-grade NMIBCs have recently been published. All patients had failed at least two courses on intravesical therapy (one contains BCG), and the preliminary analysis showed that the combination therapy had an overall response rate of 67% and an acceptable safety [60].

Two other trials are still ongoing for BCG-unresponsive patients: the Keynote-676 and the CheckMate 9UT trials. The Keynote-676 RCT is an open-label phase III study randomizing patients to receive either pembrolizumab and BCG versus BCG alone in high-risk NMIBCs, persistent or recurrent after adequate BCG induction (ineligible patients: patients who received BCG maintenance or a second induction). The primary endpoint is the CCR rate in patients with CIS, defined as the absence of high-risk NMIBC as determined by urine cytology, biopsy, radiology assessments, and local cystoscopy evaluation.

The other ongoing RCT is the CheckMate 9UT trial, which analyzes nivolumab monotherapy versus nivolumab + BMS-986205 (target therapy, IDO-1 inhibitor) with or without BCG in patients with BCG-unresponsive CIS, with or without a papillary-associated tumor. The first results are not yet available. Finally, a recent phase II RCT studied the association between avelumab and radiotherapy (whole bladder, 60–66 Gy) for BCG-unresponsive NMIBC, in patients unfit for radical cystectomy. It is a promising trial, though it is not yet recruiting.

## 5. Non-Metastatic Muscle-Invasive Bladder Cancer

### 5.1. Neoadjuvant Single-Agent Immune Checkpoint Therapy

Radical cystectomy remains the standard of care for patients with non-metastatic MIBC, and ICI therapy is not currently approved for non-metastatic MIBC. However, in patients with MIBC unfit for cisplatin-based neoadjuvant chemotherapy, the use of ICIs in the neoadjuvant setting could be favorable (Table 1). The use of pembrolizumab and atezolizumab for neoadjuvant treatment of non-metastatic MIBC has recently been tested in two phase II nonrandomized clinical trials (PURE-01 for pembrolizumab [61], and ABACUS for atezolizumab [62]).

The PURE-01 phase II clinical trial enrolled patients (*n* = 114) with T2-T3a N0 M0 bladder cancers who refused neoadjuvant chemotherapy, and who received three cycles of pembrolizumab (200 mg every three weeks) until radical cystectomy. A pathologic complete response rate (pT0 at surgery) was achieved overall in 37% of patients, ranging from a response rate of 16% in patients who had a predominant histological variant to 53% in patients with a non-predominant histological variant.

Overall, PD-L1 positivity was observed in 59% of patients (in 45% of patients with pure urothelial bladder cancer). The pathologic complete response rate depended on PD-L1 expression: 39.8% of patients with PD-L1-positive pure urothelial BC reached a complete response rate, against 25.3% of PD-L1-negative tumors.

The ABACUS phase II clinical trial evaluated the pathologic complete response rate of two cycles of atezolizumab (1200 mg every three weeks) in 95 cisplatin-ineligible MIBCs (T2-4a N0 M0). A complete response rate was overall achieved in 31% of patients (95% confidence interval, CI: 21–41%). Moreover, 17% (95% CI: 5–37%) of patients with cT3-T4 MIBCs showed a complete response rate. On the other hand, disease progression occurred in 16% of patients (95% CI: 7–27%). Once again, PD-L1 expression was crucial in predicting response to therapy: the complete response rate was 37% (95% CI: 21–55%) for patients positive for PD-L1 at baseline, while patients with PD-L1-negative tumors had a 24% complete response rate.

### 5.2. Neoadjuvant Combination Therapy: Immune Checkpoint Inhibitors + Cisplatin-Based Chemotherapy; Immune Checkpoint Inhibitor + Immune Checkpoint Inhibitor

At present, for non-metastatic MIBCs, the pathologic complete response rate (T0 at radical cystectomy) of neoadjuvant pembrolizumab/atezolizumab remains inferior to that of cisplatin-based chemotherapy, which ranges between 30% and 40% [4]. For this reason, several ongoing clinical trials are assessing the feasibility of a combination therapy (cisplatin-based chemotherapy *plus* ICIs) to gain higher response rates and disease-free survival in tumors with or without PD-L1 expression. The majority of these trials consider the pathologic downstaging to non-muscle-invasive bladder cancer at the time of radical cystectomy (T1 N0 M0) as the primary endpoint. Furthermore, preliminary results are currently only available from two phase I/II trials (Table 1).

The first one is the GU14-188 trial, which is a phase Ib/II clinical trial assessing the clinical efficacy of five preoperative cycles of pembrolizumab *plus* four cycles of gemcitabine/cisplatin on 40 patients with MIBC (T2-4a N0 M0). A total of 52% of them expressed PD-L1, and the majority of them (about 50%) presented a T2 clinical stage. A pathological downstaging at radical cystectomy (T1 N0 M0) was reached in 60% of patients, both in tumors with or without PD-L1 expression. The estimated one-year overall survival and disease-specific survival were 94% and 97%, respectively [63].

The Bladder Cancer Signal Seeking Trial BLASST-1 is a phase II clinical trial evaluating the combination of neoadjuvant nivolumab *plus* gemcitabine/cisplatin (four cycles overall) on 41 patients with MIBC T2-T4a N0-N1 M0 [64]. Preliminary data (on 29 patients) showed a pathological downstaging in about 83% of patients.

Combinations of different ICIs have recently been shown to increase clinical efficacy and pathologic complete response rates as compared to single-agent ICI treatment.

The only clinical trial with available preliminary data is the NABUCCO phase Ib single-arm trial that evaluated the combination of three neoadjuvant cycles of ipilimumab (CTLA-4 inhibitor) and nivolumab (PD-1 inhibitor) sequentially administered to 24 patients with stage III MIBC (T3-T4a N0-N1 M0), with 42% of patients presenting with regional lymph node metastasis. All patients were unfit for cisplatin chemotherapy. A pathologic complete response rate was observed in 46% of the overall patients, and in 60% of patients with PD-L1-positive tumors [65].

Other ongoing phase I/II clinical trials are evaluating nivolumab (PD-1 inhibitor) *plus* urelumab (CD137 antibody; NCT02845323 trial) or radiotherapy (RACE IT trial/NCT03529890 trial). Others are assessing combinations of durvalumab (PD-L1 inhibitor) *plus* olaparib (PARP inhibitor; NEODURVARIB trial), tremelimumab (CTLA-4 inhibitor; three trials: DUTRENEO trial, NCT03234153, and NCT02812420), or olecumab (CD73 inhibitor; BLASST-2 trial); finally, another one is combining pembrolizumab and epacostat (indoleamine 2,3-dioxygenase-1 inhibitor; PECULIAR trial/NCT03832673). The preliminary data from all these trials are not yet available. A common aspect of these trials is that they target two complementary steps in the cancer immunity cycle (see Figure 1): one step is overcoming the brake at either the T cell priming (tremelimumab) or the T cell killing phase (nivolumab, durvaluma, pembrolizumab). The combination partner either provides additional T cell stimulation (CD137 antibody) or targets the cancer cell (radiotherapy, PARP inhibitor) for potential release of new tumor antigens, or inhibitory metabolites of the TME (inhibiting CD73, which produces an immune inhibitory adenosine, epacostat, which inhibits the metabolic depletion of trypthophan).

## 6. Metastatic Muscle-Invasive Bladder Cancer

### 6.1. First-Line Immunotherapy

Up to 50% of patients with metastatic BC are ineligible for cisplatin-based chemotherapy, due to renal insufficiency, older age, or comorbidities [7]. ICIs have recently been introduced as first-line therapy for these platinum-unfit patients, with pembrolizumab (PD-1 inhibitor) and atezolizumab (PD-L1 inhibitor) being the first agents to be approved by the FDA and EMA in 2017 for patients with a positive PD-L1 status.

Several randomized phase III trials are currently investigating the role of ICIs in the first-line setting of these patients; at the moment, published data are available only from two of them, which are both single-arm phase II trials (Table 2).

The KEYNOTE-052 trial [66] evaluated pembrolizumab as first-line treatment in 370 cisplatin-unfit patients with advanced or metastatic tumors. After a median follow-up of 9.5 months, the trial showed an objective response rate (ORR) of 24%, which could be further divided into a partial response rate (the percentage of patients whose disease decreased) of 19% and a complete response rate (the percentage of patients whose disease disappeared) of 5%. Notably, the ORR was higher in patients who expressed PD-L1. The IMvigor210 trial [11] tested atezolizumab in 119 patients with the same basal characteristics. The ORR was 23%, with a 9% complete response rate. The median overall survival was 15.9 months, but the trial missed its primary endpoint of prolonging disease-free survival compared with observation only.

Published data from these two trials, even if encouraging and favorable, are challenging to interpret because of missing control arms and because of the heterogeneity of PD-L1 expression among recruited patients in both trials. On the other hand, these data seem to imply FDA recommendation in 2018 warned that patients who express low levels of PD-L1 and who receive monotherapy (atezolizumab or pembrolizumab) had decreased overall survival as compared to patients who received cisplatin-based chemotherapy.

As a consequence, these data seem to imply that patients with a negative PD-L1 status should be treated with chemotherapy-based combinations, whereas ICIs should be offered only to patients who express a high level of PD-L1.

However, due to the lack of results from other ongoing trials (CheckMate274 trial, evaluating nivolumab versus observation only, and AMBASSADOR trial, evaluating pembrolizumab versus observation only), the role of ICIs in the first-line setting of locally advanced/metastatic platinum-ineligible BC must be cautiously accepted.

### 6.2. Second-Line Immunotherapy for Platinum Pre-Treated Patients

ICIs approved by the FDA for second-line treatment of metastatic BC progressing during or after platinum-based chemotherapy are pembrolizumab, nivolumab, atezolizumab, durvalumab, and avelumab. All have demonstrated similar clinical efficacy and safety in phase I, II, and III trials (Table 2).

The randomized, open-label, phase III KEYNOTE-045 trial [8] tested pembrolizumab (either as monotherapy or in combination with paclitaxel, docetaxel, or vinflunine) on 542 patients as second-line treatment. The trial showed a significant median overall survival benefit in the pembrolizumab arm (10.3 months versus 7.4 months in the chemotherapy arm, independently of PD-L1 expression levels), a 7% complete response rate, and a 22% partial response rate.

Both 1-year and 2-year overall survival rates were higher with pembrolizumab (44% and 27%, respectively), compared to chemotherapy (30% and 14%, respectively), and pembrolizumab showed a higher rate of duration of response lasting more than 12 months (68% versus 35%).

Atezolizumab is also approved for metastatic patients who progressed during or after first-line platinum-based chemotherapy, according to the results of both phase II [15] and phase III trials [14]. The multicenter phase II NCT02108652 trial investigated the use of atezolizumab in 310 patients with inoperable locally advanced or metastatic tumors. At primary analysis, atezolizumab resulted in significantly improving the objective response rate for all patients (15%), and this rate rose up to 27% in patients with higher PD-L1 expression. According to these results, atezolizumab was approved by the FDA for the second-line therapy after chemotherapy failure. The phase III randomized control trial IMvigor211 included 931 patients comparing atezolizumab with second-line chemotherapy (paclitaxel, docetaxel, or vinflunine). Randomization was stratified by PD-L1 expression, chemotherapy type, liver metastases, and number of prognostic factors. Unlike the previous trial, atezolizumab was not associated with significantly longer overall survival than chemotherapy in patients with high PD-L1 expression (11.1 months versus 10.6 months; stratified hazard ratio, HR = 0.87; 95% confident interval, CI: 0.63–1.21, *p* = 0.41). Objective response rates were similar between treatment groups in the population with higher PD-L1 expression (≥5% of TILs): 23% in the atezolizumab group and 22% in the chemotherapy group. On the other hand, the median duration of response was longer in the atezolizumab group than in the chemotherapy group (15.9 months versus 8.3 months), and FDA maintained atezolizumab for use as second-line therapy.

A phase IV single-arm safety study trial (SAUL trial; NCT02928406) conducted on a broader population (997 patients), confirmed atezolizumab’s efficacy and tolerability profile [16].

Nivolumab was approved for second-line treatment in 2017 by the FDA, based on the results of the multicenter CheckMate275 single-arm phase II trial [46], enrolling 270 platinum-pre-treated patients. At a median follow-up of 7 months, an objective response was achieved in 19.6% of patients, and the clinical benefit was irrespective of PD-L1 expression. Moreover, overall survival was 8.7 months in the entire group.

There are two other ICIs currently FDA approved for second-line therapy. Avelumab was approved in 2017, based on the results of an open-label phase I trial [9] and the results of two expansion cohorts of the phase I trial [10]. In both studies, the objective response rate was 17%, with 6% of complete response and 11% of partial response. Once again, the response rate was higher for high-PD-L1 expression tumors. Durvalumab approval was based on a phase I/II multicenter study [67] on 191 patients with a median follow-up of 5.78 months. The objective response rate was 17.8% (including seven complete responses), earlier (median time to response: 1.41 months), and higher in high-PD-L1 expression tumors. Median progression-free survival and overall survival were 1.5 months and 18.2 months, respectively.

## 7. Immune-Related Adverse Events (irAEs)

Immune-related adverse events (irAEs) are toxicities caused by non-specific activation of the immune system, potentially affecting any organ [68]. Responses to combined ICI approaches are often better, as discussed before. However, it is still unclear if the toxicity of a single agent has a combined effect with the toxicity of the second agent.

Many irAEs are driven by the same immunologic mechanisms responsible for the therapeutic effects (i.e., blockade of inhibitory mechanisms that suppress the immune system). Skin, gut, endocrine, lung, and musculoskeletal irAEs are relatively common, whereas cardiovascular, hematologic, renal, neurologic, and ophthalmologic irAEs occur less frequently [69]. IrAEs with grade ≥3 severity occur in ≤20% of patients taking PD-1/PD-L1 agents [70,71], while the majority of irAEs are mild to moderate. However, occasionally life-threatening irAEs have been reported, with treatment-related deaths occurring in up to 2% of patients in clinical trials [72].

The clinical trial on treatment with nivolumab in patients with metastatic/surgically unresectable locally advanced bladder cancer (*n* = 270) [52] showed an incidence of grade 3–4 irAEs of 18% (48 patients), mostly fatigue and diarrhea, relatively higher than that of other ICIs (Table 2).

The irAE incidence during treatment with atezolizumab was 12% (14 patients) for all grades and 7% (8 patients) for grades 3–5 in patients without prior treatment for metastatic BC (*n* = 119). In patients with metastatic disease who progressed during platinum therapy (*n* = 310), irAE incidence was 10% (31 patients) for all grades and 6% (19 patients) for grades 3–5 (11, Table 2). The most common irAEs of grades 3–5 were transaminitis and hyperbilirubinemia. Notably, according to the data from the available published clinical trials, the incidence of serious (grades 3–5) irAEs is remarkably lower than that of chemotherapy combination regimens (see Table 2).

The real-world use of ICIs is growing and, considering that real-world patients can be frailer and have more comorbidities than patients in clinical trials, the overall amount of irAEs is expected to be higher. Research on strategies aimed at reducing the toxicity of ICIs has high clinical relevance: an interesting research field in this respect is, in the authors’ opinion, targeted delivery of drugs to the TME, such as packing ICIs into a carrier (such as a nanoparticle, or a liposome) able to release the drug selectively to the TME. Other possible strategies such as oncolytic viruses or combining vaccination and ICI treatment could become promising in the future and could be able to reduce treatment toxicity.

## 8. Critical Analysis of the Clinical Results: The Complex Immune Landscape of Bladder Cancer

Bladder cancer has a high mutational burden [73], and most cases show a high T cell infiltrate and high levels of PD-L1 expression, grouping this tumor type into the inflamed immune category [74]. While these features promise good response rates to ICI therapy, the observed low response rates indicate that other mechanisms are active. Indeed, inhibitory cells such as macrophages and Treg cells are frequently observed in BC, particularly in those with genomic alterations of EGFR, TP53, and PIK3CA [75], and higher numbers correlate with a poor prognosis [25]. Loss of MHC molecules has also been described [76], which would counteract any positive contribution towards T cell stimulation that a high TMB might have, because the required antigen presentation is impossible. Interestingly, loss and lower expression of MHC class II on tumor cells compared to a normal epithelium has been reported uniquely in BC compared to other tumor entities, suggesting that a CD4 T cell immune reaction might be more beneficial than a CD8 response [77,78]. This contention received support through a recent finding of cytotoxic CD4 T cells in BC [79,80] which expressed features of antitumor reactivity.

### A Biomarker Panel to Describe the Tissue’s Immune Status Selected from the View of Immunology

Considering the components that govern checkpoint inhibition and immune regulation, a marker panel is proposed by one of the authors (E.N.) that should allow judging the tissue’s immune status for therapy selection. Since T cells are the targets of checkpoint therapeutics, the presence of T cells must be documented, which is possible through CD3. Assessing the presence of macrophages and Treg cells, as well as PD-L1, will yield the required information about regulatory components. The marker panel should be performed by immunohistology on the tumor section and interpreted based on absolute cell numbers as well as the spatial distribution. Spatial assessment of marker distribution is an essential component allowing one to judge the status of the existing tumor immune status. The proximity of CD8 to PD-L1, for example, identifies a situation of IFN-γ-regulated adaptive PD-L1 expression, particularly if other T cell-deplete areas of the tumor are PD-L1 negative. The situation of PD-L1/CD8 proximity highlights a situation of adaptive resistance where an antitumor reactive T cell infiltrate is present and stopped by PD-L1. These tumors should have a high likelihood of response to checkpoint therapeutics [34,81,82,83,84]. Moreover, immune cell numbers should be judged as ratios. For example, high numbers of Treg cells are often found in tumors with an overall high T cell infiltration. If considering Treg cell numbers by themselves, the conclusion could be drawn that high numbers of Treg cells are good for prognosis. Seeing their numbers in relation to the overall T cell infiltrate will help to better judge the role that the cell types play in the tissue’s immune regulation.

## 9. Therapy Selection Based on the Cancer Immunobiology

The cancer immunity cycle provides a conceptual framework to assist therapy selection. The proposed biomarker panel will identify the players that are present in the tumor tissue. The best or maximally beneficial therapy or therapy combination is selected according to this information to restart and push the immunity cycle forward. Since the immune landscape may differ between patients, tumor tissue of each patient must be analyzed individually instead of judging broadly according to the tumor entity, in order to prevent selecting a patient for a likely inefficient therapy.

### 9.1. Cold Tumors Require Starting the Immune Response

While most UCB cases are classified as inflamed with a complex immune cell infiltrate, around one third of UCB cases are non-inflamed [74]. These cold tumors lack a noticeable T cell infiltrate and are predictably resistant to ICI, since T cells, which would be the targets for reactivation by checkpoint antibodies, are absent. The situation of a cold tumor indicates, immunologically, that the cancer immunity cycle was not started. Many different mutually non-exclusive factors can contribute to this situation: (i) lack of antigen release by tumor cells, (ii) inhibition of antigen uptake, antigen presentation, and maturation of DCs, (iii) inhibition of DC migration to the lymph nodes, (iv) inhibition of effective T cell priming through CTLA-4 or Treg in the lymph nodes, (v) inhibition of T cell trafficking from the lymph nodes to the tumor site. Any tumor-targeted therapy that destroys cancer cells and facilitates the release of antigens might be suitable to start the cycle (chemotherapy, radiation, molecular targeted therapies). Hypothetically, tumor-targeted therapy might be beneficially combined with DC activation, in order to enable antigen presentation, and with checkpoint inhibition to prevent the cycle from stopping, particularly if PD-L1 is expressed in the TME. There are new reagents available for DC activation (nadofaragene firadenovec, rAD-IFN-2b, an adenoviral gene transfer vector that delivers interferon-2b expression to the bladder epithelium) and for support in T cell activation (IL-15; Bempegaldesleukin, a CD122-biased agonist), which may improve the management of cold immunophenotypes [25].

Cytotoxic chemotherapy and ICI can synergistically work within the cancer immunity cycle [75]. For example, cisplatin may not only destroy tumor cells but may also deplete macrophages and suppressor myeloid cells. Thereby, it can provide antigens for T cell activation and also make the TME permissive for antitumor T cell activity [85]. Promising results have been found for the combination of pembrolizumab with an antibody–drug conjugate (enfortumab vedotin) targeting tumor-expressed nectin-4 with an antibody to release the antimicrotubule agent auristatin-E for tumor cell destruction [86].

### 9.2. T Cell-Inflamed/Hot Tumors Require Informed Decision Based on the Tumor’s Individual Immune Landscape

T cell-inflamed tumors exhibit the primary requirement for checkpoint inhibition therapy, which is the T cells. However, the “release of the brake” will only yield a response when no other hindrances are present in the TME. Therefore, patients selected for checkpoint monotherapy must have tumors with T cells in the TME, and inhibitory cells (i.e., macrophages or Treg cells) must be infrequent. T cells should be in proximity to PD-L1, which indicates adaptive resistance of an antitumor reactive T cell infiltrate that can be reactivated by “release of the brake” (as discussed above). Dual-checkpoint blockade with nivolumab plus ipilimumab can be predicted to yield a deeper response since support in T cell priming against newly released tumor antigens should occur through the activity of ipilimumab [28]. In order to support long-term T cell function and maximize the antitumor response, the immunologist suggests an “add-in” of T cell-activating agents, such as Bempegaldesleukin, a CD122-based agonist, or IL-15 [87,88]. In the TME, T cells acquire a dysfunctional state over time, which can be inhibition or exhaustion [89,90]. The latter state is the consequence of chronic TCR stimulation in a setting of persistent antigens. The expression of checkpoint molecules, including PD-1, TIGIT, TIM-3, and LAG-3, and transcription factors TCF-1 [91] and TOX [92], allows discriminating T cells with different states of exhaustion, ranging from pre-exhausted T cells with potential to be reinvigorated by checkpoint inhibitors to terminally exhausted T cells, which are no longer responsive to reinvigoration [93]. An analysis of CD8 TILs from BC identified both types of exhaustion, terminally exhausted PD-1highTOX+CD8+ TILs and PD-1+TOXlow less exhausted T cells [94]. Interestingly, terminally exhausted CD8 TILs were not responsive to single anti-PD-1 treatment in vitro but could be reinvigorated through combined checkpoint blockade with anti-PD-1 and anti-TIGIT, another checkpoint molecule that was found to be co-expressed on CD8 TILs from BC. This finding suggests co-blockade of PD-1 and TIGIT as a promising therapeutic option in BC patients. The feasibility of such a combination and improvement in tumor response has recently been documented in a phase II study in patients with non-small cell lung cancer [95].

### 9.3. Inflamed–Suppressed Tumors

For inflamed–suppressed tumors, where T cells are found together with macrophages or Treg cells, checkpoint inhibition, by itself, will not be sufficient to reactivate the T cell response. ICI needs to be combined with targeted therapies that are directed at the prevailing suppressor mechanisms. Besides chemotherapy, which is known to influence regulatory mechanisms [25,85], more specific targeting reagents are being developed. For Treg-infiltrated tumors, checkpoint inhibition may be combined with CCR4 blockade (mogamulizumab) or ipilimumab, both of which have been shown to deplete Treg cells in experimental models. For tumors with macrophage infiltration, emactuzumab (anti-CSF1R) and an LXR agonist (RGX-104) are being explored [25] (see Figure 1).

## 10. Conclusions

Immunotherapy has a role in the treatment of BC due to these tumors’ high TMB and mostly prominent immune infiltrate. However, checkpoint inhibition is not successful in all patients. The disappointing clinical results reflect the complexity of the immune landscape of BC. The therapy or combination has to be adjusted to the tumor’s immunobiology. To assess a tumor’s immunophenotype, a marker panel to identify the T cells, macrophages, and Treg cells, together with PD-L1, is suggested. Applied by immunohistology, information on the quantity of the prevailing immune cell types and their spatial information can be gained, which allows judging the immune status and therapeutic choices. Considering the complexity of the immune response, shown by the cancer immunity cycle, the antitumor response can be halted at many steps, each one requiring specific corrective measures to restart the cycle and push it forward. Checkpoint inhibition monotherapy is expected to be successful only in a minority of patients who have T cell-positive tumors and a TME permissive for T cell reactivity (no inhibitory cells). The current knowledge suggests that the immune landscape of BC is complex with the presence of T cells and multiple suppressor mechanisms, including PD-L1, macrophages, Treg cells, and loss of MHC. Therefore, combination therapies of checkpoint therapeutics with other agents that address the patient’s individual TME composition are required for most patients. Regarding the choice of combination partners for checkpoint inhibitors, seemingly “inactive” substances in monotherapy should not be excluded. In monotherapy, their activity may be concealed because the T cells’ brakes are active, but synergistic activity can be expected in combination with checkpoint inhibition. New developments are on the horizon, such as the analysis of urine lymphocytes [96], and radiolabeled antibodies or radiopharmaceuticals [97,98]. These non-invasive tools may allow dynamically assessing the BC immune TME and imaging an agent’s potential effectiveness. Non-effective therapies may then be discontinued earlier and patients transferred faster to hopefully more beneficial therapies.

## Figures and Tables

**Figure 1 cancers-13-06016-f001:**
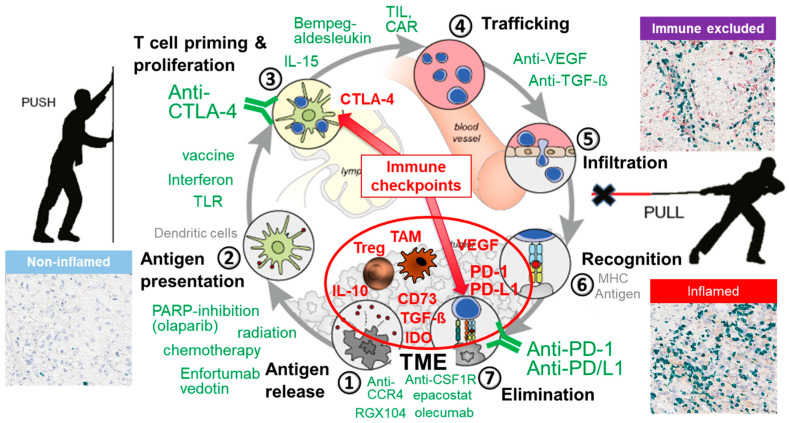
Cancer immunity cycle and therapeutic choices: 7 steps are required for a successful antitumor immune response [23]. CTLA-4 (cytotoxic T lymphocyte antigen-4) and PD-1 (programmed death-1) with its ligand PD-L1 are checkpoints—in lay terms, the physiologic brakes of the immune cycle—which control the extent of the immune response at the priming (CTLA-4) and at the elimination phase (PD-1/L1), preventing the development of too many T cells and autoreactive T cells, and collateral tissue damage, respectively. Pushing and pulling is required to keep the cycle turning until all antigens are eliminated successfully [24]. Tumors employ different strategies to interrupt the cycle, thereby escaping immune-mediated elimination. Beyond the checkpoints CTLA-4 and PD-1/L1, the tumor microenvironment (TME) is one major hurdle with many different T cell-inhibiting mechanisms, including tumor-associated macrophages (TAM), regulatory T cells (Treg), and inhibitory factors (VEGF, IL-10, TGF-ß, CD73, IDO). Where the cycle is interrupted can be retrieved from the tumor immunobiology. Immunohistology images illustrate inflamed (hot), non-inflamed (cold), and immune-excluded tumors (green represents Vina Green staining of CD3 (T cells)). In inflamed/hot tumors, T cells can be detected in the TME, indicating that steps 1–5 had occurred, and T cell activity and tumor elimination are blocked by the TME. Likewise, in the immune-excluded tumor, the cycle was started, but T cells were hindered from infiltration into the tissue and, thus, recognition and elimination cannot occur. In the non-inflamed/cold tumor, no/few T cells are detected in the TME, indicating that the cycle was not started. Therapeutic choices need to consider the interruption point and repair the defect, restarting the cycle and pushing it forward. In green, exemplary therapeutic choices are presented. Olecumab: CD73 inhibiting antibody; epacostat: indoleamine 2,3-dioxygenase-1 inhibitor; RGX 104: LXR agonist.

**Figure 2 cancers-13-06016-f002:**
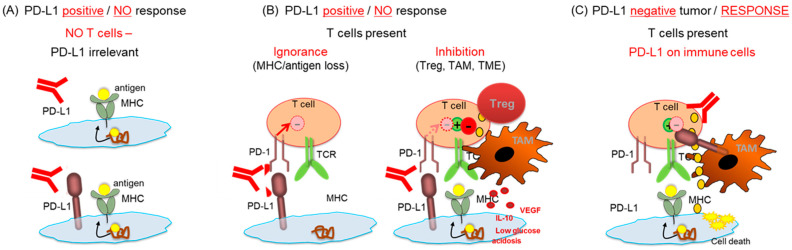
PD-L1 cannot be a sufficient predictor of response to ICI: (**A**) If T cells are not present, response to anti-PD-1//L1 therapeutics cannot occur, irrespective of PD-L1 expression. (**B**) Despite the presence of T cells and PD-L1 positivity, response might not occur if the tumor lacks MHC/antigens (T cell ignorance), or if suppressive cells (Treg, TAM) or inhibitory soluble metabolites are present (T cell inhibition). (**C**) Response might occur despite absence of PD-L1 on tumor cells if PD-L1 is expressed on other cells, i.e., macrophages, a situation which is frequent in BC.

**Figure 3 cancers-13-06016-f003:**
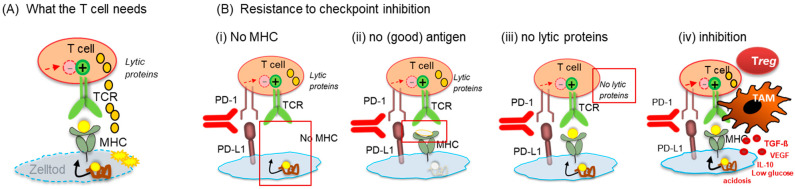
What the T cells need—prerequisites for responsiveness to ICI (**A**) and resistance mechanisms (**B**): (**A**) For the T cell to be an effective killer, it requires a positive signal through the T cell receptor (TCR), which must be provided by the tumor cell through MHC-presented antigens. If the TCR is triggered successfully, the T cell can kill the target cell if the T cell expresses lytic proteins (perforin or granzymes). (**B**) The T cell cannot be a killer (even if the PD-1/PD-L1 pathway is de-blocked by an antibody) if the tumor cell has no MHC (i), if the tumor cell has no (good) antigen (ii), if the T cell has no lytic proteins (iii), or if suppressor cells (tumor-associated macrophages (TAM) or regulatory T cells (Tregs)), or a combination of any one of these conditions, are present in the neighborhood of the T cells (iv).

**Table 1 cancers-13-06016-t001:** Neoadjuvant clinical trials.

Agent	Administration Conditions	Trial Name/NCT Number	Clinical Stage	Patients	Age (Median; IQR); Male%	PD-L1 +	Phase	Toxicity Grade 3–4 * (%)	Results
**Single-agent ICI**									
Pembrolizumab [61]	200 mg; 3 cycles, 3 weekly	PURE-01/NCT02736266	T2-3aN0M0	114	66 (60–71); 82%	59%	II	2.6%	Overall pCRR: 37% (39.8% of the PD-L1 +)
Atezolizumab [62]	75 pts: full treatment (2 cycles, 3 weekly); 20 pts: only 1 cycle	ABACUS/NCT02662309	T2-4aN0M0	95	74 (68–77); 85%	41%	II	14.7%	Overall pCRR: 31% (37% of the PD-L1 +)
**Combination therapy**									
Pembrolizumab plus Gem ± Cis [63]	Pembro: 200 mg (day 8) for 5 doses; Cis: 70 mg/m^2^ (day 1); *Gem*: 1000 mg/m^2^ (days 1 + 8), every 3 weeks for 4 cycles	GU14-188/NCT02365766	T2-4N0M0 (T2: 50%)	40	65 (−); 75%	52%	Ib/II	32.5% Grade 3–4 cytopenia: 57%	T1N0M0 at RC: 60%; 1-year OS: 94%
Nivolumab *plus* Gem ± Cis [64]	*Cis:* 70 mg/m^2^ (day 1), *Gem*: 1000 mg/m^2^ (day 1 + 8), *Nivo*: 360 mg (day 8) every 3 weeks for 4 cycles	BLASST-1/NCT03294304	T2-4aN0-1M0 (T2N0: 90%)	43	-	-	II	20%	Overall pCRR: 65.8%; downstaging: 83%
Nivolumab *plus* Ipilimumab [65]	*Ipi*: 3 mg/kg (day 1), *Ipi + Nivo*: 1 mg/kg (day 22), *Nivo*: 3 mg/kg (day 43)	NABUCCO/NCT03387761	T3-4aN0M0 or N+	24	-	60%	Ib	42%	Overall pCRR: 46%

PCRR: pathologic complete response rate; Gem: gemcitabine; Cis: cisplatin; RC: radical cystectomy; OS: overall survival; IQR: interquartile rate. * Adverse events according to the National Cancer Institute Common Terminology Criteria for Adverse Event classification, version 5.0.

**Table 2 cancers-13-06016-t002:** Immune checkpoint inhibitors for the treatment of metastatic cisplatin-unfit bladder cancer.

Agent	Administration Condition	Trial Name/NCT Number	Clinical Stage	Patients	Age (Median; IQR); Male%	PD-L1+	Phase	Toxicity Grade 3–4 * (%)	Results
**First-line therapy**									
Pembrolizumab [66]	200 mg on day 1 of each 3-week cycle, for up to 24 months	KEYNOTE-052/NCT02335424	N+: 14%; visceral M+: 85%; (liver: 21%)	370	74 (34–94); 77%	65%	II	Grade 3: 14% Grade 4: 1%	Overall ORR: 24% (CR: 5%; PR: 19%)
Atezolizumab [11]	1200 mg every 3 weeks until unacceptable toxicity or radiographic progression	IMvigor120/NCT02108652	N+: 26%; visceral M+: 66% (liver: 21%)	119	73 (51–92); 81%	67%	II	7%	Overall ORR: 23% (CR: 9%); median OS: 15.9 months
**Second-line therapy**									
Pembrolizumab versus Pacli/Doce/Vinflu [8]	*Pembro*: 200 mg every 3 weeks; *Pacli*: 75 mg/m^2^ every 3 weeks; *Vinflu*: 320 mg/m^2^ every 3 weeks, until unacceptable toxicity/radiographic progression/up to 24 months of Pembro	KEYNOTE-045/NCT02256436	-	542	Pembro: 67 (29–88); 74.1% Chemo: 65 (26–84); 74.3%	-	III	Pembro: 15%; Chemo: 49.4%	OS: 10.3 months (pembro) versus 7.4 months (chemo; *p* = 0.002); CR: 7%; PR: 22%
Atezolizumab [15]	1200 mg on day 1 of 21-day cycles, until radiographic progression/loss of clinical benefit or unmanageable toxicity	NCT02108652	N+: 14%; visceral M+: 78% (liver: 31%)	310	66 (32–91); 78%	67%	II	16%	Overall ORR: 15% (CR: 5%; PR: 10%)
Atezolizumab versus Pacli/Doce/Vinflu [14]	*Atezo*: 1200 mg on day 1 of 21-day cycles; *Vinflu*: 320 mg/m^2^; *Pacli*: 175 mg/m^2^; *Doce*: 75 mg/m^2^ on day 1 of each 21-day cycle, until disease progression/unacceptable toxicity	IMvigor211/NCT02302807	N+: 13%; visceral M+: 77% (liver: 29%)	931	Atezo: 66 (33–88); 76% Chemo: 67 (31–84); 78%	25%	III	Atezo: 6.1% Chemo: 46.5%	PD-L1+ patients: OS: 11.1 months (atezo) versus 10.6 months (chemo; *p* = 0.41); ORR: 23% (atezo) versus 22% (chemo)
Nivolumab [52]	3 mg/kg every 2 weeks until disease progression/unacceptable toxicity	CheckMate275/NCT02387996	N+: 16%; visceral M+: 84% (liver: 28%)	270	66 (38–90); 78%	30%	II	18%	Overall ORR: 19.6% (28.4% in PD-L1+); CR: 2%; PR: 17%; OS: 8.7 months
Avelumab [18]	10 mg/kg every 2 weeks until disease progression/unacceptable toxicity	JAVELIN/NCT01772004	visceral M+: 84%	249	68 (63–76); 72%	33%	Ib	8%	Overall ORR: 17%; overall CR: 6% (10% in PD-L1+); overall PS: 11% (14% in PD-L1+)
Durvalumab [67]	10 mg/kg every 2 weeks for up to 12 months or until disease progression/unacceptable toxicity	NCT01693562	N+: 7.3%; visceral M+: 93% (liver: 43%)	191	67 (34–88); 71.2%	51%	I/II	6.8%	Overall ORR: 17.8% (CR: 3.7%; PR: 14.1%); OS: 18.2 months

ORR: objective response rate; CR: complete response; PR: partial response; OS: overall survival; Pacli: paclitaxel; Doce: docetaxel; Vinflu: vinflunine: Atezo: atezolizumab; Nivo: nivolumab; Chemo: chemotherapy. * Adverse events according to the National Cancer Institute Common Terminology Criteria for Adverse Event classification, version 5.0.

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
