# Peer review of "Checkpoint Inhibition in Bladder Cancer: Clinical Expectations, Current Evidence, and Proposal of Future Strategies Based on a Tumor-Specific Immunobiological Approach"

_cancers, 2021, doi:10.3390/cancers13236016_

Round 1

Reviewer 1 Report

Mancini et al. present a good review about the role of the checkpoints inhibitors administered in bladder cancers.

This review is very detailed, well built and very interesting. The authors recall the different types of bladder cancers and the place of chemotherapies and ICI. They present the mechanisms of each ICI and the limits of their purpose. This part is really well documented and precise. The figures are clear and explain all points.

Then, the authors propose a synthesis of randomized clinical trials on immuntherapy in bladder cancers. This review will be regularly updated to keep its interest. Studies are described with their results and limits. All indications are considered and ongoing studies are also presented. Tables are clear and illustrate each study.

Finally, the authors suggest a critical analysis of the clinical results and propose some limitations of the ICI use regarding the complex immune landscape of bladder. Some perspectives of biomarkers and treatments are proposed.

This review must be published regarding minors revisions :

  • In the described studies (Tables), please indicate :
    • The administration conditions of ICI (dosing, flat dose or not, rythm)
    • Age of patients (mean, min-max)
  • Line 382-395, p 9 : describe the interest of these new targets (studied drugs in phase II/III), please.
  • Is there a position about PD-L1 expression of tumor such as a criteria to start this treatment?
  • Toxicities (G3-4) of ICI are not presented: it would be necessary to discuss, according to the different combinations of treatments (ICI alone, Double ICIs, ICI plus chemotherapy)
  • The bibliography may be revised: the order of appearance in the text is not always respected (27 and 25 line 141, p4) and some references are in text (Papaionnou legend of Figure 1, and Haslam, line 135, p4)

Author Response

We have modified the manuscript and the Tables, after careful consideration of all the reviewer N.1 comments. We have highlighted in the revised manuscript all changes made within the text.

A detailed explanation of the changes made to the manuscript can be found below.

Mancini et al. present a good review about the role of the checkpoints inhibitors administered in bladder cancers. This review is very detailed, well built and very interesting. The authors recall the different types of bladder cancers and the place of chemotherapies and ICI. They present the mechanisms of each ICI and the limits of their purpose. This part is really well documented and precise. The figures are clear and explain all points. Then, the authors propose a synthesis of randomized clinical trials on immunotherapy in bladder cancers. This review will be regularly updated to keep its interest. Studies are described with their results and limits. All indications are considered and ongoing studies are also presented. Tables are clear and illustrate each study. Finally, the authors suggest a critical analysis of the clinical results and propose some limitations of the ICI use regarding the complex immune landscape of bladder. Some perspectives of biomarkers and treatments are proposed.

This review must be published regarding minors revisions:

In the described studies (Tables), please indicate:

    • The administration conditions of ICI (dosing, flat dose or not, rythm)
    • Age of patients (mean, min-max)

We have revised the Tables I and II, and we have added two columns, one with the administration conditions of ICI and one with the age of patients.

Line 382-395, p 9: describe the interest of these new targets (studied drugs in phase II/III), please.

Is there a position about PD-L1 expression of tumor such as a criteria to start this treatment?

We have made extensive changes to the text. All the possible changes, including some speculations and original hypothesis of the authors are highlighted in yellow in the revised manuscript.

Toxicities (G3-4) of ICI are not presented: it would be necessary to discuss, according to the different combinations of treatments (ICI alone, Double ICIs, ICI plus chemotherapy)

We have added a whole new paragraph on toxicity, highlighted in yellow, putting together all the information we could get, plus some authors’ opinions and speculations on possible strategies to reduce toxicity, and ideas for future research projects

The bibliography may be revised: the order of appearance in the text is not always respected (27 and 25 line 141, p4) and some references are in text

We have carefully revised the bibliography, putting the order of appearance in the right sequence, and also adding new references related to the new points discussed in the revised text.:

We thank Reviewer N.1 for the suggestions. We feel that now the paper and tables are more interesting and updated and the bibliography is significantly improved. We hope that after these changes, the manuscript will be now found suitable for publication in Cancers.

Reviewer 2 Report

This review provides an update on the current knowledge on immune checkpoint inhibitors in the treatment of bladder cancer. The authors provide a very comprehensive and structured review with a critical analysis of the published data, including recent clinical studies.

Minor comments are listed below to improve the overall quality of the manuscript.

Minor specific comments

  • Would structure the chapter 2.2 differently and suggest to use the chapters 2.3 up to 2.6 as sub-chapters of 2.2 as they describe “biomarkers”. I mean:

2.2. Biomarkers for therapy selection: considerations from immunobiology

2.2.1 PD-1/PD-L1

2.2.2 Tumor mutational burden (TMB)

2.2.3 Tumor-infiltrating immune cells (TIL) and T cells

2.2.4 Gene expression signatures  

Additionally, I would move the chapter 2.7 “Anti-CTLA4 antibodies” in chapter 2.2.1 as it deals with ICI.

  • Maybe the authors could mention in the chapter 2.2.2 “TMB” the predictive value of TP53/PIK3CA/ATM mutations in BC (see recent work of Pan YH in Front Immunol. 2021)
  • TGFb is barely mentioned as one of the key metabolite/factor that contributes to T cell exhaustion/exclusion in the TME (see work of Mariathasan S in Nature 2018).
  • In chapter 2.2.4 “Gene signature” I would suggest to discuss briefly key transcription factors like TOX, which has been clearly shown to characterize best along with PD-1 and TIGIT the exhausted status of TiLs in BC.

Minor general comments

  • Inconsistency in the references (citation style in the text and format): for example, in Figure 1 legend, the authors cite two works: “Chen and Mellman, 2013” refers to reference 23 and the second one “Papaioannou et al. 2016” is missing in the reference list.

  • Abbreviation should be defined once throughout the text.

Author Response

We have made all the modifications required after careful consideration of all the reviewer 2 comments. We have highlighted in the revised manuscript all changes made within the text.

A detailed explanation of the changes made to the manuscript can be found below.

This review provides an update on the current knowledge on immune checkpoint inhibitors in the treatment of bladder cancer. The authors provide a very comprehensive and structured review with a critical analysis of the published data, including recent clinical studies. Minor comments are listed below to improve the overall quality of the manuscript.

Minor specific comments

  • Would structure the chapter 2.2 differently and suggest to use the chapters 2.3 up to 2.6 as sub-chapters of 2.2 as they describe “biomarkers”. I mean:

2.2. Biomarkers for therapy selection: considerations from immunobiology

2.2.1 PD-1/PD-L1

2.2.2 Tumor mutational burden (TMB)

2.2.3 Tumor-infiltrating immune cells (TIL) and T cells

2.2.4 Gene expression signatures  

Additionally, I would move the chapter 2.7 “Anti-CTLA4 antibodies” in chapter 2.2.1 as it deals with ICI.

This is an excellent suggestion and the new structure of Chapter 2 is implemented in the revised manuscript. Chapter 2.7. (anti-CTLA-4) was moved and integrated into 2.1. We thank the Reviewer for these comments.

  • Maybe the authors could mention in the chapter 2.2.2 “TMB” the predictive value of TP53/PIK3CA/ATM mutations in BC (see recent work of Pan YH in Front Immunol. 2021)

Thank you for this suggestion. This topic has been integrated in the revised version chapter 2.2.4 (Gene expression signatures and genomic mutations), as well as in chapter 7.

  • TGFb is barely mentioned as one of the key metabolites/factors that contribute to T cell exhaustion/exclusion in the TME (see work of Mariathasan S in Nature 2018).

More details on TGF-ß effects are included in chapter 2.1. These include:
- the effect of TGFß on cytotoxic T cells causing loss of Perforin (ref: Adijaya Hartana C, et al. Plos ONE (2018)
- effect of TGF-ß on T cell exclusion and attenuation of checkpoint inhibition (Mariathasan S, et al. Nature 2018 )

  • In chapter 2.2.4 “Gene signature” I would suggest to discuss briefly key transcription factors like TOX, which has been clearly shown to characterize best along with PD-1 and TIGIT the exhausted status of TiLs in BC.

A discussion of this interesting topic has been included in the revised manuscript. New references have been added.

Minor general comments

  • Inconsistency in the references (citation style in the text and format): for example, in Figure 1 legend, the authors cite two works: “Chen and Mellman, 2013” refers to reference 23 and the second one “Papaioannou et al. 2016” is missing in the reference list.

The reference list was updated with new references according to suggestions above (TGFß- Mariathasan et al., Adijaya), TOX/T cell exhaustion, NK cells, and integrating the missed reference of Papaioannou et al.

  • Abbreviation should be defined once throughout the text.

Abbreviations are defined once and then used throughout the text.

It is our belief that the manuscript, after the requested revisions, is better organized, the data is more accurate and complete, and the text is more readable. The bibliography is better organized and more complete. All the authors have reviewed and approved the final version of the manuscript.

We hope that you will find it now suitable for publication in your esteemed journal.

Reviewer 3 Report

-This is an extensive, original and well written piece of work. Authors have properly introduced concepts reviewed the literature, and discuss some of the challenges in the field.

A few recommendations for improvement.

-From L141-221. If I understood correctly, this part does not focus on bladder cancer. But it’s not always clear for the readers. Authors should “as much as” possible indicate which type of cancer/clinical setting is concerned, (associated references), or just give examples when appropriate (e.g. in melanoma, NSCLC, RCC etc….

-213-214: any comment on NK cells ?

-In the first paragraphs we see a lot of comments and cons regarding PD-L1 which is not a good marker in most cases. But at the end of the day, in the setting of bladder cancer, it seems that PD-L1 can be informative, at least not so bad in some settings, plus, it remains easy to assess in clinical practice. If not done already, authors could discuss this briefly.

-L528: they seem to be a typo in this sentence. Author should double-check (must be assess patients…)

-Author have well mentioned that one solution for more effective immunotherapies may be combinatorial approaches. Though I have not seen much discussion on the “potential toxicity” issue to manage with such combinations. I would suggest the authors challenge/discuss a bit more the issue of toxicity. Could they add on their thoughts about this aspect too? are there combinations we can expect as less toxic than others ? among the 7 steps which ones ?

Author Response

We have made all the modifications required after careful consideration of all the reviewer 3 comments. We have highlighted in the revised manuscript all changes made within the text.

A detailed explanation of the changes made to the manuscript can be found below.

This is an extensive, original and well written piece of work. Authors have properly introduced concepts, reviewed the literature, and discuss some of the challenges in the field.

A few recommendations for improvement.

-From L141-221. If I understood correctly, this part does not focus on bladder cancer. But it’s not always clear for the readers. Authors should “as much as” possible indicate which type of cancer/clinical setting is concerned, (associated references), or just give examples when appropriate (e.g. in melanoma, NSCLC, RCC etc….)

Thank you for pointing this out. Indeed, chapter 2.1 is meant to outline the general concept of cancer immunity and highlight the cancer immunity cycle as a useful guide to assess immune escape and delineate a way towards therapy selection. It is not exclusively directed towards bladder cancer. A similar thinking applies to the concept of biomarkers (Chapter 2.2). The more specific immunology of bladder cancer is addressed later ion the text. The chapter  “Therapy selection” again is not specifically directed towards bladder cancer. This is intentionally done so, as it is hypothesized that the selection of the most beneficial therapy might be, at least in part, cancer type agnostic. However, your comment is well taken. We have added examples of tumor types for which respective statements are documented or hypothesized.  All these changes are highlighted in yellow in the revised version of the manuscript.

-213-214: any comment on NK cells ?

The topic NK cells is now discussed in chapter 2.2.3, and 3 new references were added to the reference list.

-In the first paragraphs we see a lot of comments and cons regarding PD-L1 which is not a good marker in most cases. But at the end of the day, in the setting of bladder cancer, it seems that PD-L1 can be informative, at least not so bad in some settings, plus, it remains easy to assess in clinical practice. If not done already, authors could discuss this briefly.

We thank the reviewer for this comment. Indeed, in the setting of bladder cancer, PD-L1 can be informative and it is easy to use in clinical practice. PD-L1 has been found to be more informative when judged as expression of immune cells instead that of tumor cells (discussed in the text and in ref. Bellmunt J, et al., Ann Oncol. 2015)

-L528: they seem to be a typo in this sentence. Author should double-check (must be assess patients…).

This sentence was re-written to better clarify the content. It now reads: “… tumor tissue of each patient must be analyzed individually instead of judging broadly…..”

-Author have well mentioned that one solution for more effective immunotherapies may be combinatorial approaches. Though I have not seen much discussion on the “potential toxicity” issue to manage with such combinations. I would suggest the authors challenge/discuss a bit more the issue of toxicity. Could they add on their thoughts about this aspect too? are there combinations we can expect as less toxic than others ? among the 7 steps which ones ?

We thank the reviewer for this precious comment. We have added a whole paragraph on toxicity, including  some speculations  and thoughts of the authors on possible future strategies, or combination of strategies, which could help to reduce toxicity. Moreover, the toxicity data has been added to the Tables on clinical trials (Table I and II).

It is our belief that the manuscript, after the requested revisions, is more accurate, the data is more complete, and the text is more readable. All the authors have reviewed and approved the final version of the manuscript.

We hope that you will find it now suitable for publication in your esteemed journal.

Reviewer 4 Report

Re:

This submitted manuscript intends to review/update an important field of bladder cancer immunotherapy, immune checkpoint inhibitors.

Major concerns:

The manuscript looks more like a review updating checkpoint molecules and their targeting with a clinical application example in bladder tumor. It does not look like a review about checkpoints inhibition in bladder cancer. While a solid background is always welcome, there are many sections in the manuscript extensively presented which deviates the reader from the main message.

There are many incomplete/misguiding information. Some examples:

Lines 534-535: “…as they lack sufficient T cells for reactivation”. Reactivation of whom? Also implies that the cold tumors were active and needs reactivation.

Lines 535-536: “the immune system was not able to start the cycle due to either the lack of antigen or the lack of antigen presentation”. There are other factors which contribute to this, like high infiltrate of immunosuppressive cells in TME, molecules which masks the tumor from immune system (eg, GM-CSF)

Lines 557-558: “Therefore, patients selected for checkpoint monotherapy must have tumors positive for T cells”. What does tumor positive for T cells mean?

Lines 562-563: “Since T cells are likely exhausted due to their antitumor reactivity”. Exhausted T cells are the T cells which failed to be primed/activated.

Many information do not have the corresponding reference.

Author Response

We have made all the modifications required after careful consideration of all the reviewer 4 comments. The text has been extensively re-written and some parts added. We have highlighted in the revised manuscript all changes made within the text.

Responses to the Reviewer N.4 concerns, and changes made to the text, can be found below.

Major concerns:

There are many incomplete/misguiding information. Some examples:

Lines 534-535: “…as they lack sufficient T cells for reactivation”. Reactivation of whom? Also implies that the cold tumors were active and needs reactivation.

Response: “since T cells, which would be the targets for re-activation by checkpoint antibodies, are absent”. The text has been changed. All changes are highlighted in yellow in the revised manuscript.

Lines 535-536: “the immune system was not able to start the cycle due to either the lack of antigen or the lack of antigen presentation”. There are other factors which contribute to this, like high infiltrate of immunosuppressive cells in TME, molecules which masks the tumor from immune system (eg, GM-CSF)

Response: This paragraph was re-written as follows: “The situation of a cold tumor indicates, immunologically, that the cancer immunity cycle was not started. Many different mutually non-exclusive factors can contribute to this situation: i) lack of antigen release by tumor cells, ii) inhibition of antigen uptake, antigen presentation and maturation of dendritic cells, iii) inhibition of dendritic cell migration to the lymph node, iv) inhibition of effective T cell priming through CTLA-4 or Treg in the lymph node, v) inhibition of T cell trafficking from the lymph node to the tumor site”.

Lines 557-558: “Therefore, patients selected for checkpoint monotherapy must have tumors positive for T cells”. What does tumor positive for T cells mean?

Response: “Must have tumors with T cells present in the TME”.

Lines 562-563: “Since T cells are likely exhausted due to their antitumor reactivity”. Exhausted T cells are the T cells which failed to be primed/activated.

Response: We are sorry for the poor wording. No, exhausted T cells are not “failed to be primed/activated”. Priming is the process in the lymph node, where naïve T cells “see” the antigen, which fits to their TCR, for the first time (presented by dendritic cells through MHC). The priming leads to differentiation into effector cells and proliferation. Effector T cells recognize and kill tumor cells. An effector T cell can kill serially, i.e. more than one tumor cell sequentially. If there are too many tumor cells, which cannot be removed, exhaustion occurs. Exhausted T cells had been primed, they had been exposed to too many tumor cells or/and “bad” conditions.

This chapter has been extensively re-written including description of exhaustion and reinvigoration.

Many information does not have the corresponding reference.

Response: These sentences have been re-written to clarify the content and several references have been added.

We thank reviewer N.4 for the useful comments. It is our belief that the manuscript, after the requested revisions, has been significantly improved, the data is more accurate and complete, while the text is more readable.

All the authors have reviewed and approved the final version of the manuscript.

We hope that you will find it now suitable for publication in your esteemed journal.

Round 2

Reviewer 4 Report

The authors are congratulated for their updated review and also for clarifying some critical definitions. From my perspective, this manuscript is now suitable for publication.

Thank you.